# Research on the Method of Rainfall Field Retrieval Based on the Combination of Earth–Space Links and Horizontal Microwave Links

Yingcheng Zhao [ID], Xichuan Liu *[ID], Kang Pu, Jin Ye [ID] and Minghao Xian

College of Meteorology and Oceanography, National University of Defense Technology, Changsha 410073, China; zhaoyingcheng17@nudt.edu.cn (Y.Z.); pukang@nudt.edu.cn (K.P.); yejin@nudt.edu.cn (J.Y.); xiaominghao17@nudt.edu.cn (M.X.)
* Correspondence: liuxichuan17@nudt.edu.cn; Tel.: +86-134-5185-6639

**Abstract:** High-precision retrieval of rainfall over large areas is of great importance for the research of atmospheric detection and the social life. With the rapid development of communication satellite constellations and 5G communication networks, the use of widely distributed networks of earth–space links (ESLs) and horizontal microwave links (HMLs) to retrieve rainfall over large areas has great potential for obtaining high-precision rainfall fields and complementing traditional instruments of rainfall measurement. In this paper, we carry out the research of combining multiple ESLs with HMLs to retrieve rainfall fields. Firstly, a rainfall detection network for retrieving rainfall fields is built based on the atmospheric propagation model of ESL and HML. Then, the ordinary Kriging interpolation (OK) and radial basis function (RBF) neural network are applied to the reconstruction of rainfall fields. Finally, the performance of the joint network of ESLs and HMLs to retrieve rainfall fields in the area is validated. The results show that the joint network of ESLs and HMLs based on OK algorithm and RBF neural network is capable of retrieving the distribution of rain rates in different rain cells with high accuracy, and the root mean square error (RMSE) of retrieving the rain rates of real rainfall fields is lower than 0.56 mm/h, and the correlation coefficient (CC) is higher than 0.996. In addition, the CC for retrieving stratiform rainfall and convective rainfall by the joint network of ESLs and HMLs is higher than 0.949, indicating that the characteristics of the two different types of rainfall events can be accurately monitored.

**Keywords:** joint network; earth–space links (ESLs); horizontal microwave links (HMLs); rainfall field retrieval; radial basis function (RBF) neural network





## 1. Introduction

As a common weather phenomenon in the atmosphere, rainfall is closely related to human life [1]. When the distribution of rainfall in space and time is abnormal, natural disasters, such as floods and mudslides, can occur and cause life-threatening and significant economic losses to humans. In addition, the abnormal variation in rainfall reflects changes in climate [2]. Therefore, real-time and accurate monitoring of rainfall is of great significance to ensure the normal operation of society and to promote the research of atmospheric sounding science, especially the retrieval of spatial distribution of rainfall on a large scale with high accuracy and high spatial and temporal resolution [3,4].

At present, the main instruments used to measure rainfall are rain gauges, weather radars and meteorological satellites [5]. The rainfall measured by the rain gauge represents the rainfall information of the meteorological station, which cannot reflect the rainfall information of other locations outside the station because of the uneven distribution of rainfall in space [6,7]. Weather radar often obtains the echoes of cloud and parts of the precipitation in the atmosphere, which is significantly different from the rainfall on the surface [8]. Moreover, the meteorological rain satellite can retrieve the spatial distribution

of a wide range of rainfall, but its spatial resolution is very low because of the long detection distance [9]. The use of a large number of rain gauges and weather radars to form a high-density meteorological observation network enables the acquisition on the distribution of rainfall over a large area with high precision and high temporal and spatial resolution [6,10]. However, the current distribution of rain gauges and weather radars in different regions is uneven, limited by complex maintenance technology and high cost. In particular, the distribution is more sparse in remote areas [11]. Therefore, the proposal of advanced rainfall detection methods to supplement the existing detection instruments to obtain better rainfall information is still a widely undertaken research topic in recent years.

With the rapid development of communication technology, a new method of retrieving rainfall by using the attenuation caused by rainfall on microwave communication links widely distributed in space has been proposed in recent years, and it has received widespread attention [12–15]. Compared with traditional rainfall measurement methods, the denser distribution of microwave links promises to enable the retrieval of rainfall over large areas. Additionally, directly using the microwave links of the existing communication base stations for rainfall retrieval does not entail too much cost, so it has the advantage of low cost [16,17]. First of all, Messer et al. [12] conducted an experiment to retrieve rainfall using microwave links in commercial cellular networks in 2006. Additionally, the experimental results showed that the performance of microwave links for rainfall measurement was better than weather radar and showed the preliminary potential of using microwave links for rainfall measurement. Then, some scholars optimized the calculation model of retrieving rainfall through the accurate modeling of microwave propagation in the atmosphere, so as to further improve the accuracy of this method [17–20]. Moreover, the technology of retrieving rainfall using microwave link is also applied to radar parameter calibration, precipitation classification and raindrop size distribution (DSD) inversion [21,22]. In addition, with the rapid development of the internet of things technology and 5G communication networks, the distribution of microwave links is becoming denser and denser [19,23]. A growing number of studies began to focus on using widely distributed horizontal microwave links (HMLs) to retrieve rainfall fields that represent the spatial distribution of rain rates in the area [24,25]. Computer tomography (CT) and inverse distance weighting (IDW) interpolation have also been applied to rainfall field reconstruction based on the microwave link network, and the results also verify the feasibility of the scheme [26,27]. However, further research results show that it is difficult for CT and IDW to reconstruct the rainfall field with high precision and high spatial resolution for the real distributed microwave links, which is affected by the link sparsity and irregular structure distribution.

Similar to HMLs, the inclined earth–space links (ESLs) between earth communication satellites and ground receiving antennas will also produce significant attenuation due to rainfall in the process of atmospheric propagation. Therefore, the method of using ESLs to retrieve rainfall has also been proposed and deeply studied recently [13,27,28]. Since 2010, a large number of experiments have begun to verify the use of a single earth–space link (ESL) to measure rainfall. Barthès et al. [13] and Mugnai et al. [27] studied the feasibility of using the attenuation of ESLs signals in Ku and Ka bands to measure rainfall, respectively. After that, many advanced techniques have been proposed to solve the key factors that affect rainfall retrieval by ESL, such as the identification of the rainy periods, reference baseline and rainfall height [28–30]. On the basis of considering the influence of microwave attenuation characteristics of the melting layer on ESL propagation, Xian et al. [28,31] improved the rain attenuation calculation model and proposed an ESL rainfall retrieval method based on machine-learning technique. In addition, the performance of the single ESL to retrieve different grades of rainfall and different types of rainfall has been verified by long-term empirical data [32]. In addition to the research on retrieving rainfall by a single ESL, Huang et al. [33] and Xian et al. [34] studied the feasibility of ESLs network retrieval of large-scale rainfall field through simulation experiments. Compared with other detection instruments, the cost of ESL equipment is lower, and the installation is more convenient.

It mainly consists of a small television antenna and a receiver at the user terminal [35,36]. Moreover, there are a large number of broadcast users all over the world. Therefore, the use of widely distributed ESLs to obtain rainfall information in mountainous areas, islands and plateaus that lack professional rainfall detection instruments is of great potential. On the other hand, with the rapid development and application of satellite communication, a large number of communication satellites will be launched for networking in the near future, such as Starlink, Oneweb and Kuiper constellations [37]. This will make a large number of ESLs closely cover all regions of the world and then providing conditions for using ESLs network to retrieve large-scale and high-precision rainfall fields.

Previous studies have always focused on ESL and HML separately. However, the use of these two new detection techniques to retrieve rainfall has the same essential physical mechanism, which is the phenomenon of microwave attenuation caused by rainfall. The realization of joint detection of the two technologies is of great significance to make full use of space-based and ground-based microwave signal resources and supplement traditional detection methods. Moreover, joint detection is more likely to obtain a large-scale rainfall field with higher accuracy and higher resolution. In this paper, we carry out the research of using the joint network of ESLs and HMLs to retrieve rainfall field with high precision and high resolution. Firstly, the simulation experiment of joint network of ESLs and HMLs is carried out based on the atmospheric propagation model of ESL and HML, and a rainfall detection network for retrieving rainfall field is built, which is composed of multiple ESLs and HMLs. Then, the sparse rainfall distribution obtained by rainfall detection network is reconstructed by using ordinary Kriging interpolation (OK) and radial basis function (RBF) neural network. As a result, the continuous distribution of rainfall fields in the area is obtained. Finally, the method is verified by the hybrid rain cell (HYCELL) model and the real rainfall fields. The purpose of this work is to preliminarily verify the feasibility of using ESLs combined with HMLs to retrieve rainfall field and to provide reference for future applications.

In this article, the major contributions are as follows.

(1) A method for detecting rainfall by combining multiple sources of microwave links is proposed. We built a rainfall detection network for retrieving rainfall fields in combination with ESLs and HMLs, and we validated the significant potential of the method to retrieve high-precision rainfall fields using the HYCELL model and real rainfall fields.

(2) The OK algorithm and RBF neural network are applied to the joint network to retrieve rainfall fields. The results indicate that the joint networks of ESLs and HMLs based on the OK algorithm and RBF neural network can both retrieve the distribution characteristics of rainfall accurately. Moreover, the overall performance of the RBF neural network is better than that of the OK algorithm.

## 2. Principles of Rainfall Field Retrieval by ESLs and HMLs

Microwave signals will have significant attenuation due to scattering and absorption of rainfall. This physical phenomenon can lead to serious rain attenuation of the radio wave signals from communication satellites to the earth and the communication links between ground-based communication base stations due to rainfall. The ingenuity of the new technique of retrieving rainfall by using rain attenuation from ESL and HML is to reverse the interference of rainfall in the atmosphere to microwave signals to obtain rainfall information. In addition, ESLs and HMLs can be combined to retrieve rainfall field when multiple ESLs and HMLs are present in the area. Figure 1 depicts the process of retrieving rainfall field by combining ESLs and HMLs. Firstly, a link network for rainfall observation can be built by a combination of ESLs and HMLs in the area. Then, the sparse rain attenuation at the location of the link network can be obtained by using the forward model of rain attenuation for ESL and HML. Furthermore, the inversion model of microwave rain attenuation can be used to convert the rain attenuation into sparsely

distributed rain rates in the area. Finally, the spatial reconstruction method can be used to obtain the continuously distributed rainfall field.

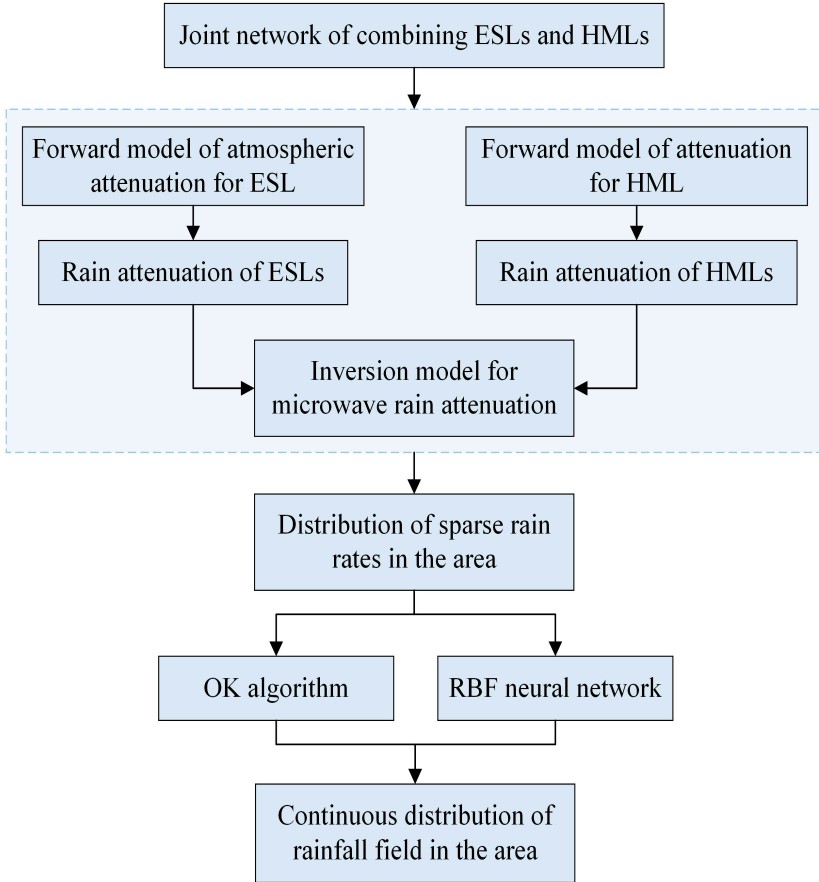

**Figure 1.** The process for retrieving rainfall fields in the area by combining ESLs and HMLs.

### 2.1. Principle of Rainfall Retrieval by ESL

The basic principle of using ESL to retrieve rainfall is briefly described as follows. When rainfall occurs in the ESL between the satellite and the antenna, the microwave signal of the ESL will be weakened due to the effect of scattering and absorption. Then, the changing process of microwave signal transmission can be monitored by the receiver of the ground satellite antenna. At this time, the microwave rain attenuation can be obtained by using the corresponding extraction algorithm. Finally, the average rain rate on the path of ESL can be obtained by the calculation model between microwave rain attenuation and rainfall [38]. Next, a detailed analysis of the radiation transfer process of ESL is presented.

After passing through the atmosphere, the microwave signal sent by the satellite reaches the ground satellite antenna and is transmitted to the receiver for processing. We use $P_t$ and $P_r$ to represent the transmitting power of the satellite and the receiving power of the ground receiver, respectively, and have the following equation [32]:

$$P_r(t) = P_t(t) \cdot G \cdot \eta_A \cdot \eta_F \cdot \eta_R + P_n(t) \tag{1}$$

where $G$ and $P_n$ represent the antenna gain and the noise power of the signal, respectively. $\eta_A$ represents the attenuation coefficient of non-rainfall factors in the atmosphere, mainly related to clouds, gas composition and atmospheric turbulence. Additionally, $\eta_F$ and $\eta_R$ represent the attenuation coefficients of ESL in free space and rainfall, respectively. In

addition, the signal-to-noise ratio (SNR) of the ESL signal received by the ground receiver can be expressed as

$$\text{SNR}(t) = 10\log\frac{P_t(t){\cdot}G{\cdot}\eta_A{\cdot}\eta_F{\cdot}\eta_R}{P_n(t)}. \tag{2}$$

The free space loss $A_F$ and the path-integrated rain attenuation $A_R$ of ESL are defined as

$$\begin{cases} A_F(t) = -10\log(\eta_F) \\ A_R(t) = -10\log(\eta_R) \end{cases}, \tag{3}$$

then, Equation (2) can be written as

$$\text{SNR}(t) = 10\log\frac{P_t(t){\cdot}G{\cdot}\eta_A}{P_n(t)} - A_R(t) - A_F(t). \tag{4}$$

For noise power $P_n$, it mainly includes sky noise $T_{sky}$ and system noise $T_{sys}$, which can be given by

$$P_n(t) = k_B{\cdot}B{\cdot}\left(T_{sys}(t) + T_{sky}(t)\right), \tag{5}$$

where $k_B$ is the Boltzman constant, and $B$ represents the bandwidth of the ESL signal. $T_{sys}$ can be regarded as constant when the receiver of ESL works stably. In addition, when there is rainy weather, the $T_{sky}$ received by the receiver mainly comes from the influence of rainfall. Additionally, it can be obtained by

$$T_{sky}(t) = \frac{\int G(t,\Omega){\cdot}t_m(1-10)^{-\frac{A_R(t)}{10}}d\Omega}{\int G(t,\Omega)d\Omega}, \tag{6}$$

where $G(t, \Omega)$ is the antenna gain pattern at the fixed solid angle $\Omega$, and $t_m$ is the average temperature of the path [39].

We transform and rewrite Equation (5) as

$$P_n(t) = k_B{\cdot}B{\cdot}T_{sys}(t){\cdot}\frac{\left(T_{sys}(t) + T_{sky}(t)\right)}{T_{sys}(t)}, \tag{7}$$

and define

$$\begin{cases} F_n(t) = k_B{\cdot}B{\cdot}\left(\frac{T_{sys}(t)+T_{sky}(t)}{T_{sys}(t)}\right) \\ P_0(t) = k_B{\cdot}B{\cdot}T_{sys}(t) \\ V(t) = 10\log\frac{P_t(t){\cdot}G{\cdot}\eta_A}{P_0(t)} \end{cases}, \tag{8}$$

then, Equation (4) can be rewritten as

$$\text{SNR}(t) = V(t) - F_n(t) - A_R(t) - A_F(t). \tag{9}$$

When there is no rain, the $\text{SNR}_{no\text{-}rain}$ received by the ESL receiver can be regarded as the SNR caused by factors other than rainfall. Additionally, $\text{SNR}_{no\text{-}rain}$ can be written as

$$\text{SNR}_{no-rain} = V(t) - F_n(t) - A_F(t). \tag{10}$$

In order to extract the rain attenuation caused by rainfall, we use $\text{SNR}_{no\text{-}rain}$ as the attenuation baseline. Therefore, according to Equations (9) and (10), the rain attenuation $A_R$ can be rewritten as

$$A_R(t) = \text{SNR}_{no-rain} - \text{SNR}. \tag{11}$$

The rain attenuation $A_R$ can be measured at the receiver terminal of ESL by the method described above. In addition, according to the ITU-R model [40,41], the relationship between rain specific attenuation $\gamma_R$ (dB/km) and rain attenuation is obtained by

$$\begin{cases} \gamma_R = \alpha R^\beta \\ A_R = \gamma_R \cdot h_0 + 0.36/\sin\theta \end{cases}, \tag{12}$$

where $h_0$ is the height of atmospheric $0°$C-Layer, $\theta$ is the elevation of ESL, and the values of coefficients $\alpha$ and $\beta$ are obtained by ITU-R P .838-3 [40]. Therefore, combined with Equations (11) and (12), the average rain rate $R$ (mm/h) retrieved by ESL can be written as

$$R = \left[ \frac{(\text{SNR}_{no-rain} - \text{SNR})\sin\theta}{(h_0 + 0.36)\alpha} \right]^{\beta^{-1}}. \tag{13}$$

In order to further verify the corresponding relationship between SNR and rain rate, we used the ESL at Ku band built in Nanjing, China, to record the change process of SNR during rainfall. A rainfall event in June 2020 is selected to analyze the correlation between ESL signal and rain intensity. As shown in Figure 2, the SNR decreases significantly when rainfall occurs. This change becomes more obvious as the rain rate increases. In addition, the SNR can fluctuate rapidly due to noise, even when no rainfall is present. This is mainly due to the movement of the atmosphere. However, on the whole, rainfall is the main factor leading to the decrease in SNR of ESL. Further calculation shows that the correlation coefficient (CC) between SNR and rain rate is $-0.612$. This indicates that there is a strong negative correlation between the two and validates the feasibility of using ESL to retrieve rainfall.

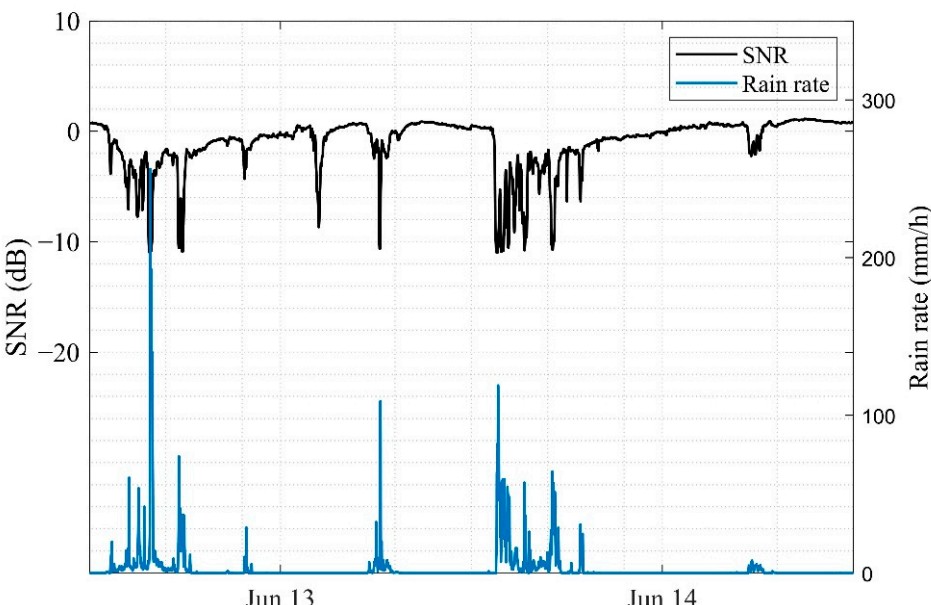

**Figure 2.** Correspondence between ESL signal and rain rate from 13 June to 14 June 2020. The black line represents the SNR of ESL, and the blue line represents the rain rate.

### 2.2. Principle of Rainfall Retrieval by HML

The principle of using HML for rainfall inversion is similar to that of ESL. It also uses the physical phenomenon that rainfall can cause attenuation of microwave signals. However, because the line-of-sight HML composed of microwave transmitter and receiver propagates near the ground, the change of meteorological elements in its path range is small. Additionally, the existing calculation models all assume that the meteorological elements at the height of HML are constant. Therefore, using HML to retrieve rainfall is

more convenient than using ESL to retrieve rainfall. Next, the process of retrieving rainfall by HML is described in detail [42].

When the microwave signal sent by the microwave transmitter of HML propagates in the atmosphere near the ground, the total power attenuation $A_{total}$ on the path can be expressed as

$$A_{total} = A_{gas} + A_{water} + A_{FS} + A_{rain} \qquad (14)$$

where $A_{gas}$ is the attenuation caused by gases (mainly oxygen) in the atmosphere, $A_{water}$ is the attenuation due to water vapor, $A_{FS}$ is the loss of microwave transmission in the path of free space, and $A_{rain}$ is the attenuation caused by rainfall. The key point in using HML to retrieve rainfall is to record rain attenuation accurately. Therefore, we rewrite Equation (14) as

$$A_{total} = A_{rain} + A_{baseline}. \qquad (15)$$

The $A_{baseline}$ represents the attenuation of the signal caused by other factors except rainfall, and it is generally approximated by the attenuation during non-rainfall periods after revision. Thus, as far as the receive signal level (RSL) of HML receiver is concerned, regarding $A_{baseline}$ as $RSL_{baseline}$, the rain attenuation can be given by

$$A_{rain} = RSL_{baseline} - RSL. \qquad (16)$$

To further describe the relationship between rain rate and rain attenuation, a model for calculating HML rain attenuation was fitted based on the Mie scattering theory and a large amount of observational data [43]. As shown in

$$A_{rain} = aR^b \cdot l, \qquad (17)$$

where $l$ is the length of HML, and the values of coefficients $a$ and $b$ are obtained by ITU-R P .838-3 [40]. Revising the coefficients $a$ and $b$ based on DSD data from different locations will further improve the accuracy of this calculation model and thus better match the practical situation in different locations. Therefore, according to Equations (16) and (17), the average rain rate retrieved by HML can be written as

$$R = \left( \frac{RSL_{baseline} - RSL}{a \cdot l} \right)^{\frac{1}{b}}. \qquad (18)$$

To visualize the correspondence between the HML signal and the rain rate, we used the HML (26 GHz, vertical polarization) set up in Jiangyin, China, to realistically record the RSL of its receiver during rainfall. Additionally, a rainfall event in June 2021 is selected to validate the correlation between RSL and rain rate as shown in Figure 3. Similar to the ESL signal, the HML signal intensity decreases significantly in the presence of rainfall. This phenomenon becomes more obvious as the rain rate increases. In addition, rainfall is the most powerful factor in the variation of the HML signal. Additionally, the CC between RSL and rain intensity was calculated to be −0.762 by further analysis. The strong correlation indicates that the HML signal is highly sensitive to rain rate and is more suitable for retrieving rainfall.

### 2.3. Rainfall Field Retrieval by Combined ESLs and HMLs

When multiple ESLs and HMLs exist within an area, the rain rates $R(x_i, y_i)$ at the location $(x_i, y_i)$ of the ESLs and HMLs can be measured. The rain rates at these different locations reflect part of the distribution of the rain rates $R(x, y)$ over the whole area. Therefore, we can combine the ESLs and HMLs in the area to obtain highly accurate rainfall fields by means of the reconstruction algorithm. In this paper, the OK algorithm and RBF neural network are applied to accomplish the rainfall field retrieval.

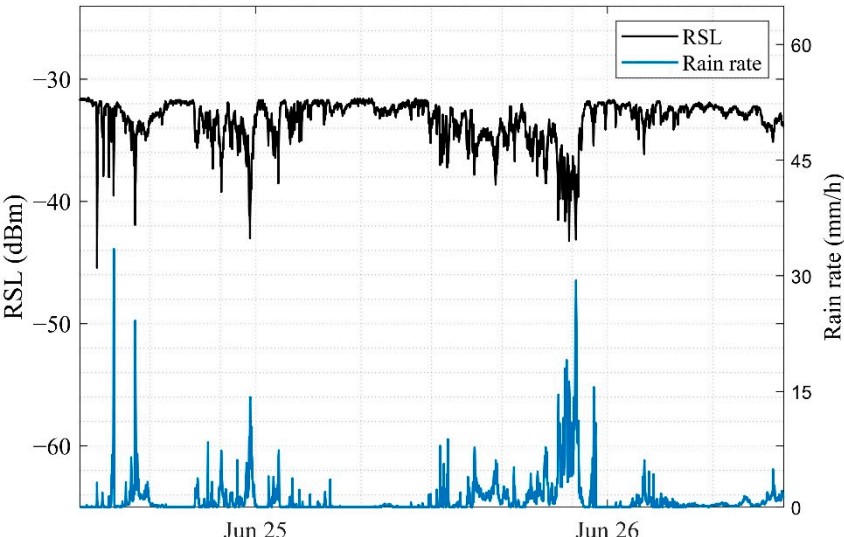

**Figure 3.** Correspondence between the HML signal and rainfall from 25 June to 26 June 2021. The black line is RSL of HML, and the blue line is rain rate.

2.3.1. Rainfall Field Reconstruction by OK Algorithm

The assumptions for reconstructing the rainfall field using the OK algorithm are as follows: (1) the rain rates at all locations in the area are interrelated; (2) the shorter the distances, the higher the correlation between the rain rates [44]. Therefore, using $R(x_i, y_i)$ from ESLs and HMLs to reconstruct the $R(x, y)$ of the whole rainfall field can be given by

$$\begin{cases} R(x,y) = \sum_{i=1}^{N} \lambda_i R(x_i, y_i) \\ \sum_{i=1}^{N} \lambda_i = 1 \end{cases}, \tag{19}$$

where $N$ denotes the number of ESLs and HMLs, and $\lambda_i$ is a weight factor indicating the degree of contribution of $R(x_i, y_i)$ to $R(x, y)$. In addition, the semi-variogram $\psi$ is defined by the mathematic expectation of the observed rain rates $R(x_i, y_i)$ and the target point rain rates $R(x, y)$, as given by

$$\psi(i) = \frac{1}{2} E\Big[ (R(x,y) - R(x_i, y_i))^2 \Big]. \tag{20}$$

Additionally, the connection between weight factor $\lambda$ and the semi-variogram $\psi$ is given by [45]

$$\begin{bmatrix} \psi_{11} & \psi_{12} & \cdots & \psi_{1N} & 1 \\ \psi_{21} & \psi_{22} & \cdots & \psi_{2N} & 1 \\ \vdots & \vdots & \vdots & \vdots & \vdots \\ \psi_{N1} & \psi_{N2} & \cdots & \psi_{NN} & 1 \\ 1 & 1 & 1 & 1 & 1 \end{bmatrix} \begin{bmatrix} \lambda_1 \\ \lambda_2 \\ \vdots \\ \lambda_N \\ \delta \end{bmatrix} = \begin{bmatrix} \psi_1 \\ \psi_2 \\ \vdots \\ \psi_N \\ 1 \end{bmatrix}, \tag{21}$$

where $\delta$ is the Lagrange multiplier. Furthermore, the semi-variogram takes the stable model for which the performance was validated in [45], and it is given by

$$\psi = A \cdot \Big( 1 - \exp\Big( -(d/B)^C \Big) \Big), \tag{22}$$

where $d$ represents the distance between the observed rain rates $R(x_i, y_i)$ and the target point rain rates $R(x, y)$, and $A$, $B$ and $C$ are coefficients, which can be estimated by fitting the observed values of ESLs and HMLs. Next, after obtaining the weight factor $\lambda$ on the basis

of the semi-variogram, we can implement the reconstruction of the rainfall field according to Equation (19).

### 2.3.2. Rainfall Field Reconstruction by RBF Neural Network

RBF neural network is a forward propagation network with good performance, which is composed of input layer, hidden layer and output layer. Among them, the transformation from the input layer to the hidden layer is nonlinear, while the transformation from the hidden layer to the output layer is linear [46]. Each unit in the hidden layer of the RBF network can be regarded as a RBF, which is a function that depends only on the distance to the origin. The hidden nodes of the RBF neural network use the distance between the input vector and the center vector as the independent variable of the function and use the RBF as the activation function. Additionally, the farther the input of the neuron is from the RBF center, the lower the activation degree of the neuron is. Thus, the approximation of the objective function only depends on the nearest RBFs, while the RBFs with long distances basically do not work. This local approximation feature enables RBF neural network to have good convergence speed.

For the reconstruction of rainfall field in the area, the rainfall, at a point in space, has the highest correlation with the nearest rainfall to it. Additionally, this distribution, characteristic of the rainfall field, coincides with the local approximation feature of the RBF neural network. Therefore, we can design the RBF neural network to reconstruct the rainfall field as shown in Figure 4. The geometric meaning is that a continuous two-dimensional rainfall field $R(x, y)$ is recovered from the sparsely distributed rain rates $R(x_i, y_i)$ retrieved from the ESLs and HMLs. The specific reconstruction function can be expressed as

$$R(x,y) = \sum_{n=1}^{m} \sum_{i=1}^{N} W_n f_n(R(x_i, y_i)),$$ (23)

where $m$ is the number of hidden units, $W_n$ denotes the weights from the hidden units to the output layer, and $f_n$ is the hidden layer RBF.

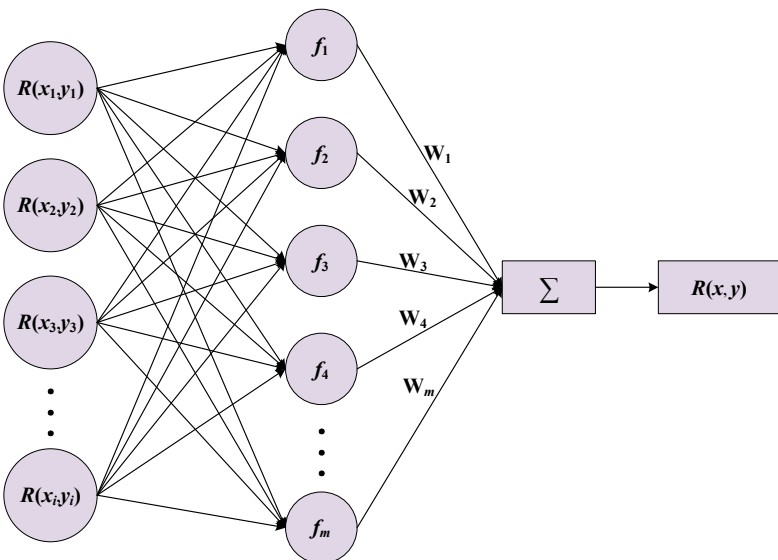

**Figure 4.** Architectural diagram of RBF neural network for retrieving rainfall fields.

In this paper, the detailed setup of the RBF neural network used to reconstruct the rainfall field is as follows:

(1) The Gaussian function is used as the hidden layer RBF and is given by

$$f_n(R(x_i, y_i)) = exp(-\frac{\|R(x_i, y_i) - C_n\|}{2\sigma_n^2}),$$ (24)

where $C_n$ denotes the center of the RBF in the hidden units, and $\sigma_n$ is the width of the hidden units, which is primarily used to adjust the resolution of the neural network.

(2) The K-means clustering is used to determine the network cluster center $C_n$ and the width of the Gaussian function $\sigma_n$.

(3) The learning rate of the RBF neural network is set to 0.01, and the minimum error requirement of the objective function and the maximum number of iterations are set to 0.01 mm/h and 5000 times, respectively. The training will be stopped automatically if the minimum error requirement is reached during the training process.

(4) The RBF neural network can automatically add the number of hidden units until the training error requirement is reached.

## 3. Design of the Joint Network of ESLs and HMLs

To verify the feasibility of combining ESLs and HMLs for rainfall field retrieval in the area, a joint network design of combining ESLs and HMLs in Xiamen, China, is carried out in this paper through simulation experiment. The latitude and longitude ranges of the experimental area are 118 °E–118.35 °E and 24.35 °N–24.7 °N. In addition, the whole experimental area is divided into a grid of 35 km × 35 km. It is assumed that the rainfall distribution within each grid point is uniform, so the spatial resolution of the rainfall field is $1 \times 1$ km$^2$. The annual average of the local atmospheric 0°C-Layer height is 4.476 km [41]. For the design of the link network, we made use of the communication links between the satellite antennas on the ground and the geostationary satellites to build ESLs, and the communication links between the microwave transmitters and the receivers on the ground to build HMLs. Additionally, the parameters of the five geostationary satellites selected for building the ESLs are shown in Table 1.

**Table 1.** The parameters of the geostationary satellites for building the ESLs.

| Satellites | Longitude | Elevation (°) | Azimuth (°) | Frequency (GHz) |
|------------|-----------|---------------|-------------|-----------------|
| Apstar9    | 142.0°E   | 50.86         | 133.13      | 11.154          |
| Apstar6C   | 134.0°E   | 56.28         | 145.56      | 12.323          |
| AsiaSat9   | 122.0°E   | 61.04         | 170.75      | 12.726          |
| ChinaSat10 | 110.5°E   | 60.10         | 197.94      | 12.309          |
| AsiaSat5   | 100.5°E   | 55.19         | 217.51      | 12.460          |

In the previous study [34], we only randomly designed the ESLs without considering the impact of a more realistic environment. Obviously, it is impossible to install the ESL antennas for measuring rainfall on the surface of the river and the sea. Therefore, the following guidelines need to be followed in the design of the joint network of ESLs and HMLs in this paper.

(1) The links for ESLs and HMLs should be spread as evenly as possible across the area.

(2) The rain rate measured by the link represents the average rain rate over the path and can be considered as the rain rate at the location of the midpoint of the link.

(3) Although rain attenuation is more likely to occur for ESLs and HMLs with long distances, the distribution of real rainfall over the area is not uniform. Thus, the long links have poor spatial representation. To improve the spatial representation of rainfall retrieved by ESLs and HMLs, it is necessary to build short links (the link lengths in this experiment are in the range of 2.8–7.6 km).

(4) The antennas of ESLs, transmitters and receivers of HMLs cannot be installed on the water surface. However, the links can pass above the water surface.

In Figure 5, we combined ESLs and HMLs to build a link network for rainfall observation, which consists of 40 links. Specifically, the 20 purple lines represent the HMLs, and the blue "o" indicates the location of the satellite antennas, which connect the communication satellites in Table 1 to form the 20 ESLs. The transmitters and receivers of the HMLs are located across the shore in order to monitor the rainfall on the water surface. In addition,

the rainfall retrieved by the link can be considered as the rainfall at the red midpoint due to the fact that it only reflects the path-averaged rainfall. To avoid excessive differences in each ESL, the ESLs were built at the Ku band and with horizontal polarization. Moreover, the electromagnetic wave signals of the HMLs are all at 20 GHz with horizontal polarization.

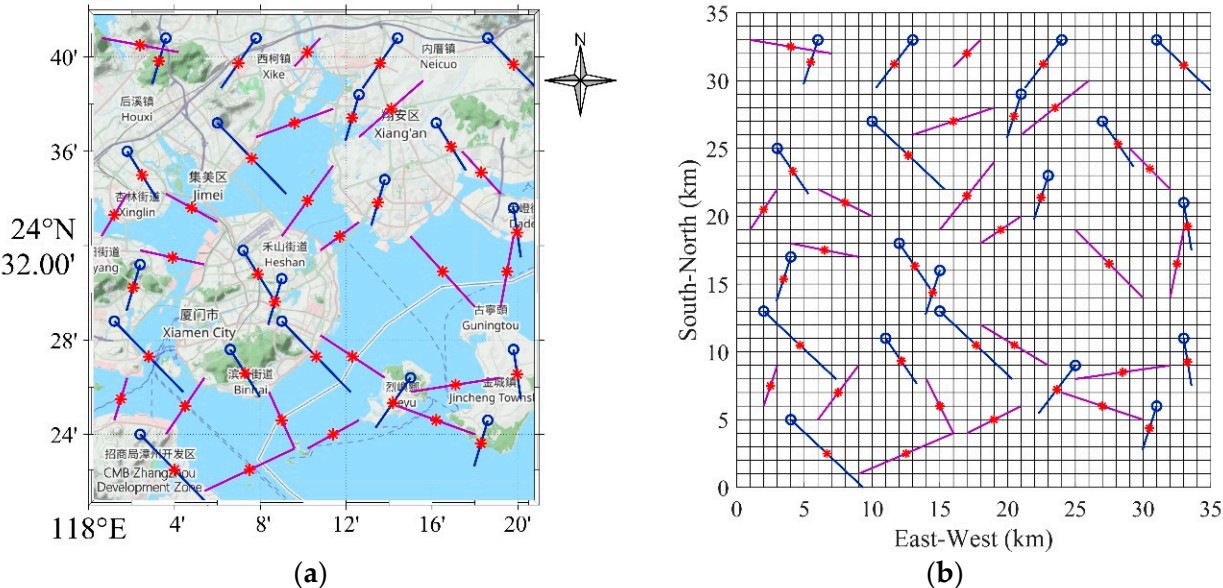

(**a**) (**b**)

**Figure 5.** Network distribution of ESLs and HMLs in Xiamen: (**a**) Distribution of the original link network; (**b**) Distribution of the link network in the gridded area. The blue lines represent the ESLs, the blue markers "o" indicate the location of the satellite antennas, the purple lines represent the HMLs, and the red markers "*" indicate the midpoint of the links.

## 4. Results and Discussion

The rainfall observation network built with multiple ESLs and HMLs enables measuring sparsely distributed rain rates in the area. Then, the OK algorithm and RBF neural network are used to reconstruct the sparsely distributed rain rates to obtain a continuous distribution of rainfall fields in the area. To investigate the performance of the link network in retrieving rainfall field, we used the HYCELL model and real rainfall fields for relevant validation. The root mean square error (RMSE), mean bias (MB) and Pearson correlation coefficient (CC) are used to give a quantitative evaluation of the retrieval results. For clarification, RMSE is mainly used to give the error between the retrieved rainfall field $R_{retrieve}$ and the real rainfall field $R_{real}$, MB gives the unbiasedness between $R_{retrieve}$ and $R_{real}$, and CC is used to evaluate the degree of similarity between $R_{retrieve}$ and $R_{real}$. Additionally, they are defined by

$$\text{RMSE} = \sqrt{\frac{1}{n}\sum_{i=1}^{n}(R_{retrieve,i} - R_{real,i})^2}, \tag{25}$$

$$\text{MB} = \frac{1}{n}\sum_{i=1}^{n}(R_{retrieve,i} - R_{real,i}), \tag{26}$$

and

$$\text{CC} = \frac{cov(R_{retrieve,i}, R_{real,i})}{\sigma_{R_{retrieve}} \cdot \sigma_{R_{real}}}, \tag{27}$$

where $cov(R_{retrieve}, R_{real})$ denotes the covariance between $R_{retrieve}$ and $R_{real}$, and $\sigma_{Rretrieve}$ and $\sigma_{Rreal}$ are the standard deviations of $R_{retrieve}$ and $R_{real}$, respectively.

*4.1. Rain Cell Retrieval by Network of ESLs and HMLs*

The "rain cell" is a meteorological term used to represent the distribution of rain rate along the path. By studying the spatial distribution characteristics of rainfall, the exponential rain cell (EXCELL) model, Gaussian model and HYCELL model are mainly used to represent the rain cell at present [47–49]. In this case, the HYCELL model with better spatial representation, which is based on a combination of the EXCELL model and the Gaussian model [50]. Additionally, the expression of the model is

$$R(x,y) = \begin{cases} R_E \cdot exp\left[ -\left( \frac{x^2}{a_E{}^2} + \frac{y^2}{b_E{}^2} \right)^{\frac{1}{2}} \right], & (R_1 < R < R_2) \\ R_G \cdot exp\left[ -\left( \frac{x^2}{a_G{}^2} + \frac{y^2}{b_G{}^2} \right) \right], & (R > R_2) \end{cases} \tag{28}$$

where $R_E$ and $R_G$ are the peak rain rate in the EXCELL model and Gaussian model, respectively, $a_E$ and $b_E$ are parameters of the EXCELL model, and $a_G$ and $b_G$ are parameters of the Gaussian model. From the model, it can be seen that when the rain rate $R$ is lower than $R_2$, the rainfall conforms to the EXCELL model. Additionally, when the rain rate is higher than $R_2$, the rainfall conforms to the Gaussian model.

To test the performance of rain cell retrieval by the network of ESLs and HMLs, we first designed various kinds of different rain cell samples in the experimental area as the initial rainfall fields according to the HYCELL model. The rain cells consist of 1225 (35 × 35) rain rate signals and thus have a spatial resolution of $1 \times 1$ km². Then, the sparsely distributed rain cells in the experimental area are retrieved by the joint network of ESLs and HMLs. Finally, the OK algorithm and RBF neural network are used to reconstruct the sparse rain cells, respectively, so as to obtain the complete rain cells in the experimental area. For the RBF neural network, we simulate 100 different rain cells as the training set according to the parameter range in Table 2, so as to determine the parameters of the neural network model. In addition, in order to verify the performance of OK algorithm and RBF neural network for reconstructing rain cells, a set of different rain cells consisting of three rain cells were designed as the validation samples of rainfall field, and their spatial distribution of rainfall is shown in Figure 6. There are significant differences in the range of rain rates and the location of peak rainfall between these rain cells.

**Table 2.** The range of parameters of the HYCELL model.

| Rain Rate | Parameters | Range |
|---|---|---|
| | $R_E$ | 10–100 mm/h |
| $R_1 < R < R_2$ | $a_E$ | 0.5–35 km |
| | $b_E$ | 0.5–35 km |
| | $R_G$ | 10–80 mm/h |
| $R > R_2$ | $a_G$ | 0.5–35 km |
| | $b_G$ | 0.5–35 km |

In the whole retrieval process, the rain rates at the location of the links in the rain cell can first be measured using the joint network of ESLs and HMLs. Then, the OK algorithm and RBF neural network are used to reconstruct the sparse rain rates measured by the network, respectively. The distribution of the complete rain rates of the rain cell is thus obtained, and the retrieval of the rain cell in the area is finally accomplished. Figure 7 shows the results of the joint network of ESLs and HMLs based on the OK algorithm and RBF neural network to retrieve the different rain cells. It can be seen from the figure that the retrieved rain cells of the OK algorithm and RBF neural network show good consistency with the initial rain cells. This indicates that these two methods can both accurately retrieve the distribution of rain rates for each of the rain cells. It should be noted that the rain rates of the rain cells retrieved using the OK algorithm are locally closer to those of the initial

rain cells. However, there are significant differences in the overall structure of the rain cells. Most notably, the retrieved results for rain cell 2 and rain cell 3 show obvious distortions in local areas compared to the initial rain cells. The reason for this is mainly due to the fact that the distribution of links at these locations is too sparse and hence causes more significant errors in the OK algorithm during the interpolation process. In addition, the structural distribution of the rain cells retrieved by the RBF neural network is generally consistent with the structure of the initial rain cells on the whole. However, the rain rate peak at the center of rain cell 2 is not accurately retrieved.

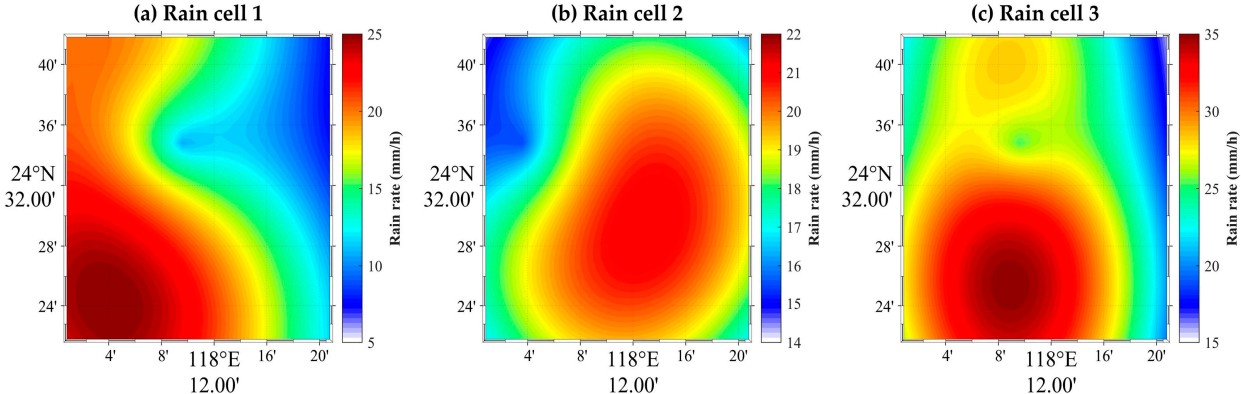

**Figure 6.** The set of different rain cells in the area. (**a**) is rain cell 1, indicating that the rain cluster has not yet fully entered the area. (**b**) is rain cell 2, indicating that the rain cluster has fully moved into the area. Additionally, (**c**) is rain cell 3, indicating that the rain cluster has split into two centers of rain rate.

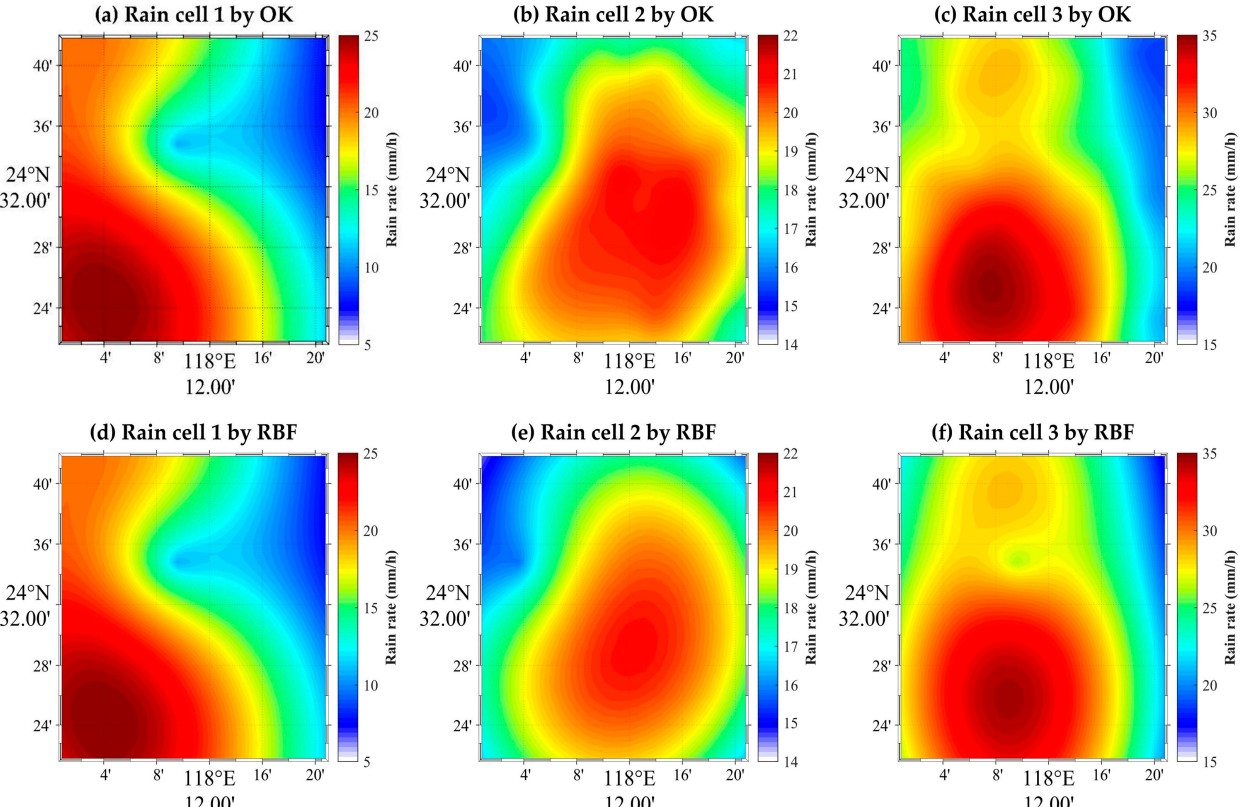

**Figure 7.** The results of the joint network of ESLs and HMLs retrieving the set of different rain cells. (**a**–**c**) represent the rain cells reconstructed by the OK algorithm, and (**d**–**f**) represent the rain cells reconstructed by RBF neural network.

To further evaluate the retrieval results quantitatively, the rain rates of the rain cell retrieved by the OK algorithm and the RBF neural network are compared with the rain rate signals of the 1225 grid points of the initial rain cell. In Figure 8, it can be observed that the rain rate signals retrieved by the OK algorithm and the RBF neural network are overall close to those of the initial rain cells. However, there are still significant deviations for the retrieval of a small number of rain rate signals. As can be seen from Table 3, the RMSE of the OK algorithm for retrieving rain cell 1 and rain cell 2 is lower than the results of the RBF neural network, which indicates that the accuracy of the OK algorithm for retrieving the rain rate signals of rain cell 1 and rain cell 2 is better than that of the RBF neural network. However, for rain cell 3, the results of the RBF neural network outperformed the OK algorithm. For the MB of retrieval results, the OK algorithm is essentially unbiased for rain cell 2 and rain cell 3, but it underestimates rain cell 1. Additionally, the RBF neural network underestimates rain cell 1 and rain cell 2, and it overestimates rain cell 3. In addition, the retrieved rain cells of the RBF neural network all have higher CC than that retrieved by the OK algorithm, which indicates that the similarity between the rain cells retrieved by the RBF neural network and the initial rain cells is better. It is further verified that the distribution of the structures of the rain cells retrieved by the RBF neural network in Figure 7 is in better consistency with the initial rain cells in general.

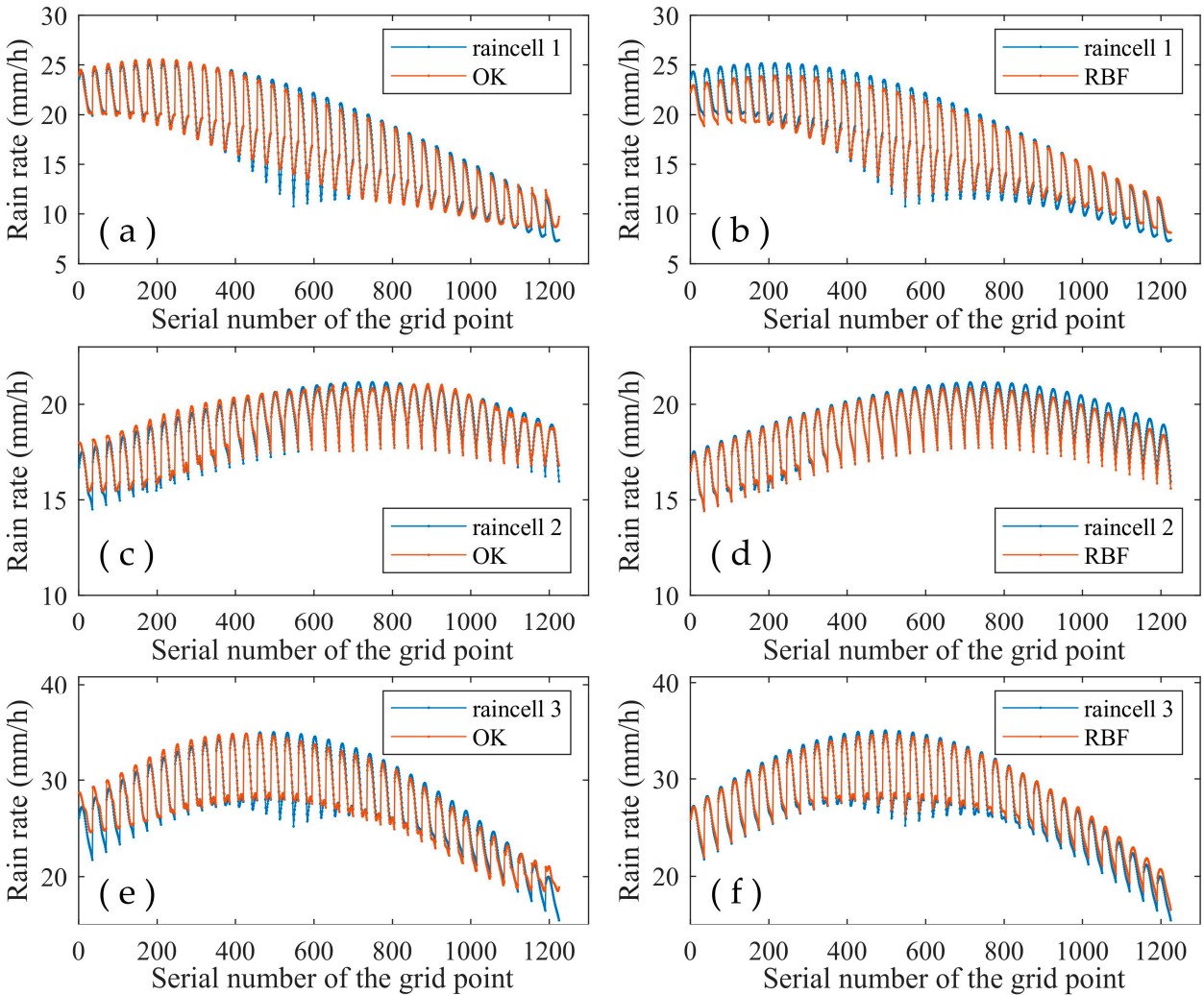

**Figure 8.** The results of joint net of ESLs and HMLs retrieving the rain rates of the different rain cells. (**a**,**c**,**e**) are the rain rates of the different rain cells retrieved by the OK algorithm, and (**b**,**d**,**f**) are the rain rates of the different rain cells retrieved by the RBF neural network.

**Table 3.** The results of the joint network of ESLs and HMLs retrieving the different rain cells.

| Rain Cells | RMSE (mm/h) | | MB (mm/h) | | CC | |
|---|---|---|---|---|---|---|
| | OK | RBF | OK | RBF | OK | RBF |
| Rain cell 1 | 0.52 | 0.69 | −0.14 | −0.12 | 0.995 | 0.999 |
| Rain cell 2 | 0.27 | 0.28 | 0.03 | −0.20 | 0.986 | 0.994 |
| Rain cell 3 | 0.75 | 0.54 | 0.02 | 0.29 | 0.985 | 0.998 |

Based on the above results, it can be seen that the RMSE and MB of the OK algorithm for retrieval of the different rain cells are lower than 0.75 mm/h and 0.14 mm/h, respectively, and the CC of the retrieved results is above 0.985. In contrast, the RMSE and MB of the RBF neural network are lower than 0.69 mm/h and 0.29 mm/h, respectively, and the CC of the retrieved rain cells is higher than 0.994. Moreover, the structural distribution of the rain cells retrieved by the RBF neural network is generally in better consistency with the initial rain cells. Furthermore, the results indicate that after the distribution of sparse rain rates in the area is measured by the joint network of ESLs and HMLs, the distribution of rain cells in the area can be accurately retrieved by the OK algorithm and the RBF neural network. Additionally, this illustrates the ability of the joint network of ESLs and HMLs to retrieve rain cells with high accuracy.

### 4.2. Performance of Retrieving Real Rainfall Field

The joint network of ESLs and HMLs is able to accurately retrieve the distribution of rain rates in rain cells over the area. However, the spatial distribution of real rainfall is more complex and irregular than the structure of simulated rain cells. Therefore, it is necessary to validate the performance of retrieving real rainfall fields using the joint network of ESLs and HMLs by real rainfall fields that match the characteristics of actual rainfall. In this paper, we constructed the real rainfall fields in the experimental area by processing the rainfall products from the Climate Prediction Center Morphing Technique of NOAA (CMORPH) through spatial interpolation, so as to further investigate the performance of retrieving real rainfall fields by using the joint network of ESLs and HMLs. Rainfall fields with a time interval of 0.5 h can be obtained through the CMORPH rainfall product, which means the time resolution of the real rainfall fields in this research is 0.5 h [51]. We chose a total of 1000 rainfall fields for the period from 2017 to 2019, each of which consisted of 1225 rainfall rates.

To verify the performance of retrieving the real rainfall field using the joint network of ESLs and HMLs, the real rainfall field from CMORPH is firstly used as the initial field, and then, the sparse distributed rain rates at the location of the links in the area can be measured by the joint network of ESLs and HMLs. Eventually, the OK algorithm and the RBF neural network are used to reconstruct the sparse rain rates, respectively, so as to obtain the rainfall field of the whole area. For the RBF neural network, we used 70% of the real rainfall fields as the training set, thus determining the parameters of the computational model for the RBF neural network to retrieve the real rainfall field. The remaining rainfall fields would be used as the initial fields to test the retrieval performance of the joint network of ESLs and HMLs.

The results of the joint network of ESLs and HMLs retrieving all rain rates from 300 real rainfall fields are shown in Figure 9. From the validation in Figure 9a, it can be seen that the RMSE and CC between the results retrieved from the network of ESLs and HMLs based on the OK algorithm and the rain rates of real rainfall fields are approximately 0.56 mm/h and 0.996, respectively, and the average rain rate retrieved is very close to that of real rainfall fields. In addition, from the validation in Figure 9b it, can be shown that the RMSE and CC between the results retrieved from the network of ESLs and HMLs based on RBF neural networks and the rain rates of the real rainfall fields are about 0.51 mm/h and 0.997, respectively, and the average rain rate retrieved is significantly lower than that of the real rainfall fields, which is more obvious during heavy rainfall. The retrieval results of the OK algorithm and the RBF neural network are generally unbiased, as evidenced by the MB,

but the RMSE and CC of the RBF neural network for retrieving the rain rates of rainfall fields are better than the retrieval results of the OK algorithm. However, the performance of the RBF neural network in retrieving the average rain rate of the real rainfall fields is not as good as that of the OK algorithm. In general, the results of the OK algorithm retrieval have a few results with large errors across the range of rain rates. This is directly reflected in the fact that some of the data points deviate from the fitting straight line for all ranges of rain rates retrieved by the OK algorithm in Figure 9a. In addition, the results retrieved by the RBF neural network are in overall better agreement with the rain rates of the real rainfall fields than those retrieved by the OK algorithm, as can be seen in Figure 9. However, the rain rates over 50 mm/h retrieved by the RBF neural network are lower than those of the real rainfall field in Figure 9b, which shows that the RBF neural network underestimates the rain rates for retrieving extreme rainfall (over 50 mm/h).

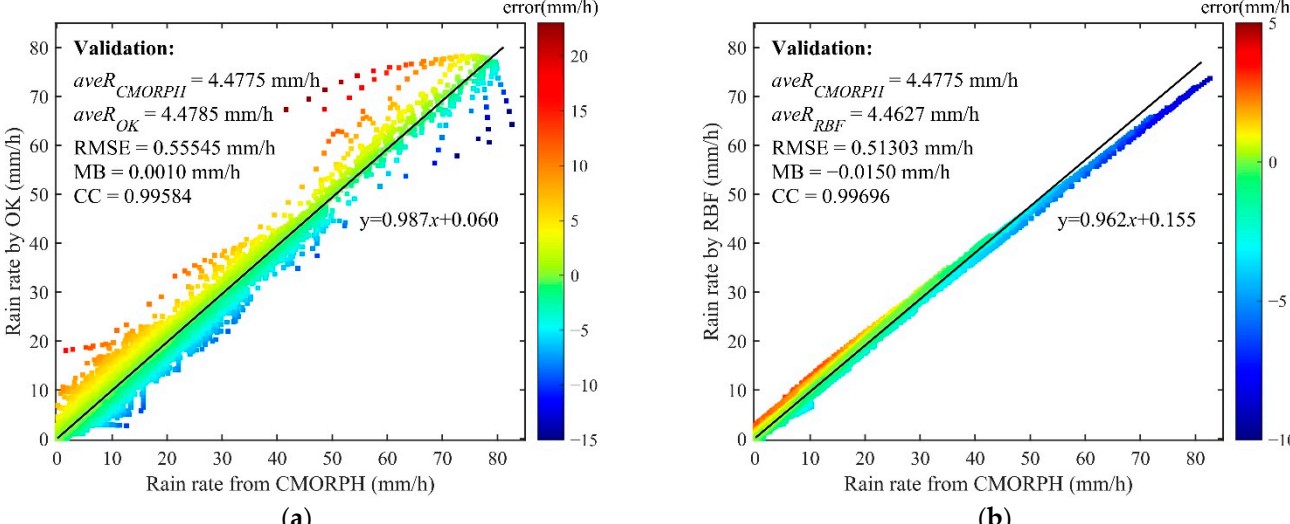

**Figure 9.** The results of the joint network of ESLs and HMLs retrieving all rain rates from real rainfall fields: (**a**) Comparison of rain rates retrieved by the joint network of ESLs and HMLs based on the OK algorithm with the real rain rates; (**b**) Comparison of rain rates retrieved by the joint net of ESLs and HMLs based on the RBF neural network with the real rain rates.

Figure 10 shows the probability density function (PDF) and cumulative distribution function (CDF) of the rain rates from all rainfall fields in the experimental area. It can be shown that extreme rainfall accounts for a small percentage of the total rainfall in the area after the statistics of all rain rates. The PDF shows that the proportion of all rain rates above 30 mm/h is below 1.0%. In addition, 99.4% of the rain rates in the sample are below 30 mm/h, as can be seen from the CDF. Therefore, the main reason for the RBF neural network to underestimate the rain rates of extreme rainfall is that the number of extreme rainfall samples in the training set is too small. However, the RBF neural network still has high accuracy for the retrieval of common rainfall in the area.

To further examine the performance in monitoring actual rainfall event, we retrieved two different types of rainfall in July 2019 using the joint network of ESLs and HMLs, respectively. Figures 11 and 12 show the results of the joint network of ESLs and HMLs retrieving the two different types of rainfall events, respectively. Among them, the rainfall field 1 to rainfall field 4 is a partial process of stratiform rainfall event with a duration of two hours. Additionally, the rainfall field 5 to rainfall field 8 is a convective rainfall event with a duration of two hours. The stratiform rainfall is characterized by small rain rate, long duration and slow movement. In contrast, the convective cloud rainfall has a high peak of rain rate, develops rapidly, and the movement of the rain cluster is fast. From monitoring the results of the stratiform rainfall and convective rainfall, it can be seen that the rainfall fields retrieved by the joint network of ESLs and HMLs based on the OK algorithm and

RBF neural network are generally consistent with the distribution of the actual rainfall. Moreover, the joint network of ESLs and HMLs can accurately retrieve the characteristics of the stratiform rainfall and convective rainfall. However, there are still differences between the retrieved rainfall fields and the structure of the actual rainfall in a small, localized area.

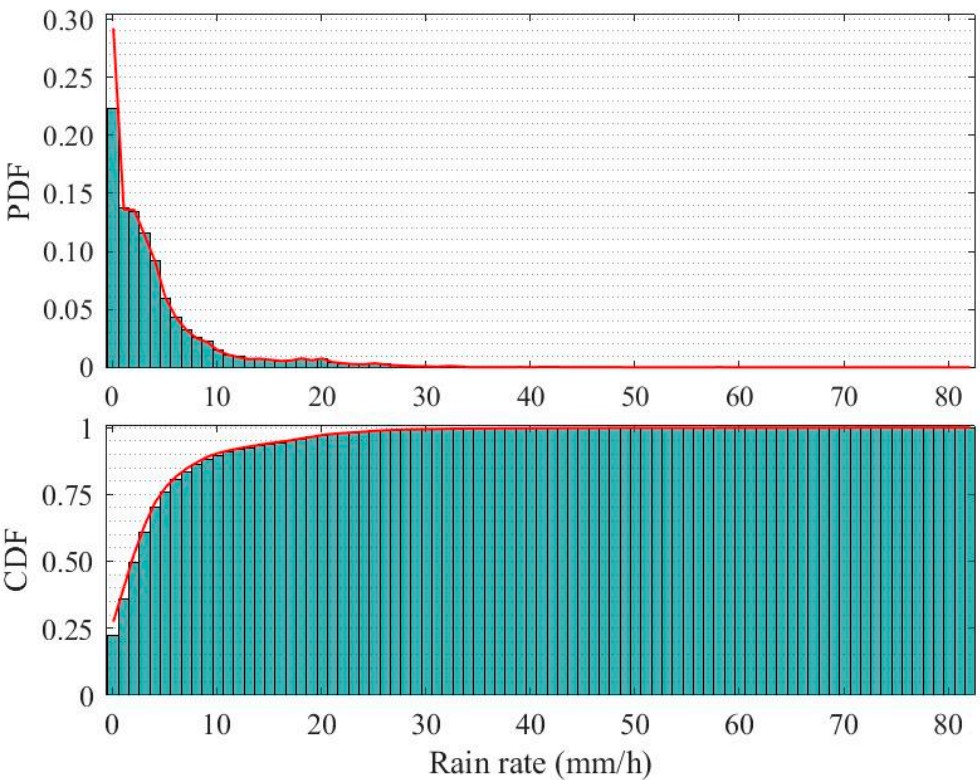

**Figure 10.** Statistics on the distribution of rain rates from all rainfall fields in the experimental area.

The RMSE, MB and CC for the two types of rainfall retrieved by the joint network of ESLs and HMLs are shown in Table 4. It can be concluded that the RMSE of retrieving stratiform rainfall and convective rainfall by the joint network of ESLs and HMLs based on the OK algorithm is lower than 0.54 mm/h and 4.04 mm/h, respectively, while the CC of retrieving stratiform rainfall and convective rainfall is higher than 0.949 and 0.971, respectively. In addition, the RMSE for retrieving stratiform rainfall and convective rainfall by the joint network of ESLs and HMLs based on the RBF neural network is lower than 0.32 mm/h and 3.44 mm/h, respectively, and the CC for retrieving stratiform rainfall and convective rainfall is higher than 0.954 and 0.997, respectively. In addition, it is revealed from the MB of the retrieval results that both the OK algorithm and the RBF neural network are essentially unbiased for the retrieval of stratiform rainfall. However, for the retrieval of convective rainfall, the OK algorithm significantly overestimates rainfall field 7, and the RBF neural network obviously underestimates rainfall field 6 and rainfall field 7, which is mainly due to the factor that RBF will underestimate extreme rainfall. Overall, the errors and unbiasedness of retrieving stratiform rainfall for the OK algorithm and RBF neural network are better than those of retrieving convective cloud rainfall, but the consistency of retrieving convective rainfall is better than the results of retrieving stratiform rainfall. Moreover, the retrieval results of the RBF neural network for both types of rainfall are generally superior to those of the OK algorithm.

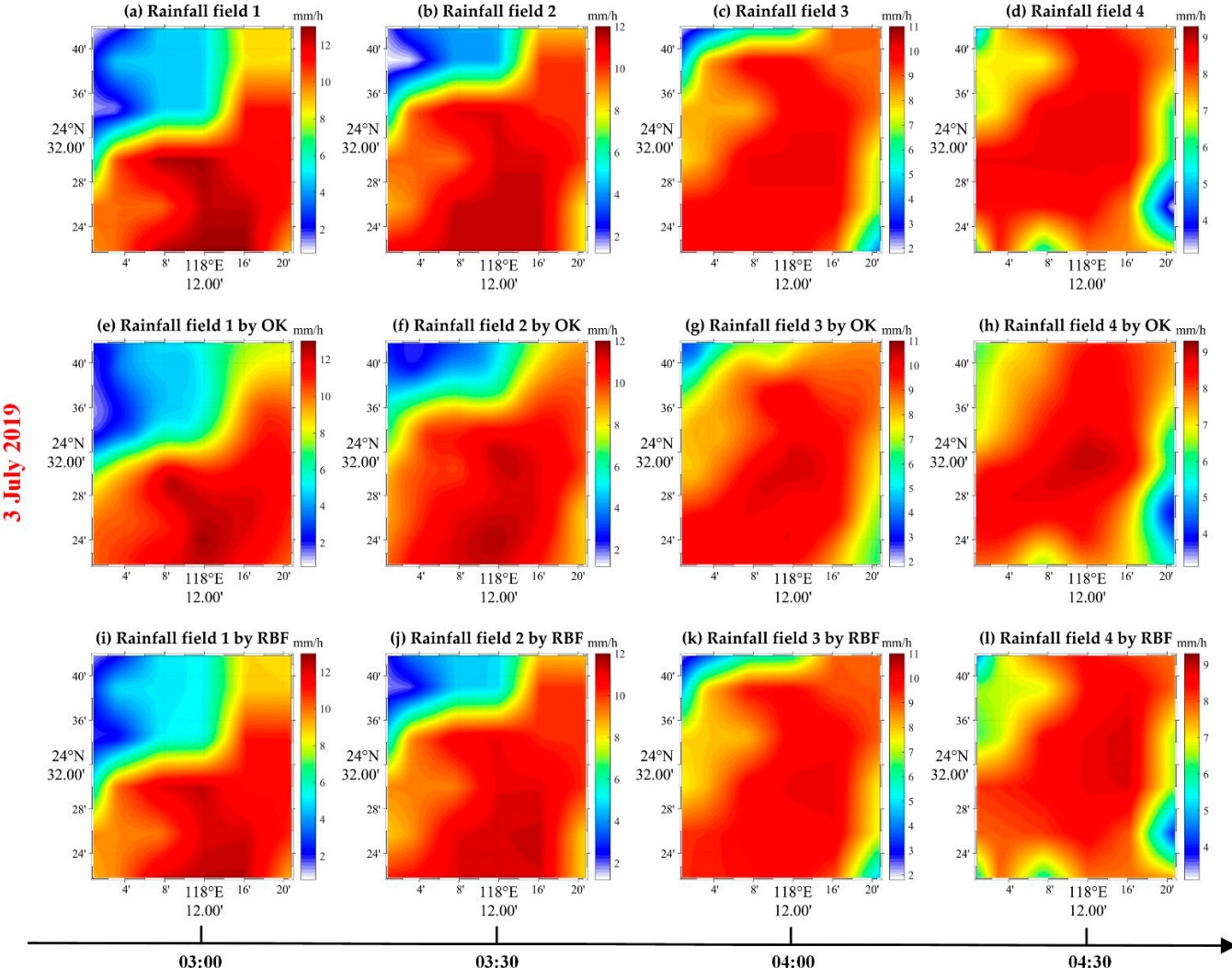

**Figure 11.** The results of retrieving stratiform rainfall event that occurred on 3 July 2019 by combining ESLs and HMLs. Rainfall field 1 to rainfall field 4 in (**a**–**d**) show the development of the stratiform rainfall at 0.5 h intervals, characterized by small rain rate, long duration and slow movement. (**e**–**h**) are the results of retrieving the stratiform rainfall from rainfall field 1 to rainfall field 4 by the OK algorithm, and (**i**–**l**) are the results of retrieving the stratiform rainfall from rainfall field 1 to rainfall field 4 by the RBF neural network.

The results of the retrieval of real rainfall fields show that the joint network of ESLs and HMLs can accurately retrieve the rain rates of real rainfall fields and can effectively monitor stratiform rainfall and convective rainfall. Thereby, the feasibility of the joint network of ESLs and HMLs to retrieve rainfall fields in the area is validated. Additionally, it also shows the significant potential of combined ESLs and HMLs for monitoring rainfall over large areas.

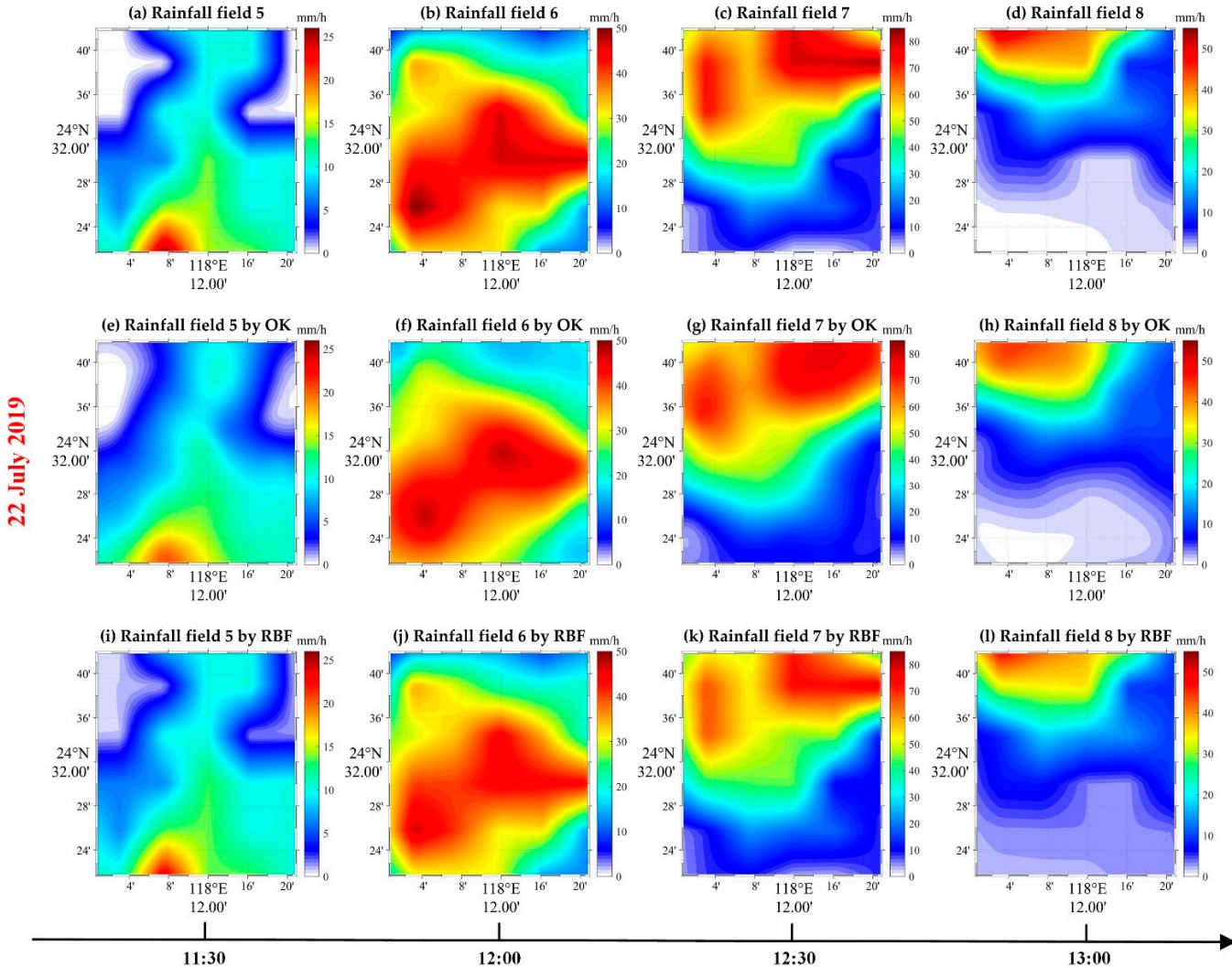

**Figure 12.** The results of retrieving convective rainfall event that occurred on 22 July 2019 by combining ESLs and HMLs. Rainfall field 5 to rainfall field 8 in (**a–d**) show the development of the convective rainfall with a high peak of rain rate, rapid development and fast movement of the rain cluster. (**e–h**) are the results of retrieving the convective rainfall from rainfall field 5 to rainfall field 8 by the OK algorithm, and (**i–l**) are the results of retrieving the convective rainfall from rainfall field 5 to rainfall field 8 by the RBF neural network.

**Table 4.** The results of retrieving two different types of rainfall events by the joint network of ESLs and HMLs.

| Rainfall Type | Rainfall Fields | RMSE (mm/h) | | MB (mm/h) | | CC | |
|---|---|---|---|---|---|---|---|
| | | OK | RBF | OK | RBF | OK | RBF |
| Stratiform rainfall | Rainfall field 1 | 0.52 | 0.32 | −0.08 | −0.01 | 0.988 | 0.999 |
| | Rainfall field 2 | 0.54 | 0.26 | −0.03 | −0.02 | 0.980 | 0.998 |
| | Rainfall field 3 | 0.42 | 0.23 | −0.01 | −0.04 | 0.964 | 0.990 |
| | Rainfall field 4 | 0.31 | 0.30 | −0.05 | −0.06 | 0.949 | 0.954 |
| Convective rainfall | Rainfall field 5 | 1.05 | 0.58 | −0.20 | 0.09 | 0.979 | 0.997 |
| | Rainfall field 6 | 2.56 | 1.60 | 0.22 | −1.00 | 0.971 | 0.997 |
| | Rainfall field 7 | 4.04 | 3.44 | 1.27 | −1.94 | 0.988 | 0.998 |
| | Rainfall field 8 | 1.75 | 1.73 | 0.47 | 0.62 | 0.992 | 0.998 |

## 5. Conclusions

In this paper, we carried out the research of combining multiple ESLs with HMLs to retrieve high-precision rainfall fields. First of all, we deployed a rainfall detection network for retrieving rainfall fields in Xiamen based on the atmospheric propagation model of ESL and HML. Then, the methods of the OK algorithm and the RBF neural network were given to reconstruct the sparse distribution of rainfall measured by the rainfall detection network to obtain a continuous distribution of rainfall fields in the area. Finally, the HYCELL model and real rainfall fields were used to validate the performance of the joint network of ESLs and HMLs to retrieve rainfall fields in the area. The experimental results verify the feasibility of combining ESLs and HMLs to accurately retrieve rainfall fields. Detailed conclusions are presented below.

(1) For the HYCELL model, the joint network of ESLS and HMLs is able to retrieve the distribution of rain rates in rain cells with high accuracy. The RMSE and MB of the OK algorithm for retrieval of the different rain cells are lower than 0.75 mm/h and 0.14 mm/h, respectively, and the CC of the retrieved results is above 0.985. In contrast, the RMSE and MB of the RBF neural network are lower than 0.69 mm/h and 0.29 mm/h, respectively, and the CC of the retrieved rain cells is higher than 0.994. Moreover, the structural distribution of the rain cells retrieved by the RBF neural network is generally in better consistency with the initial rain cells.

(2) For the rainfall from CMORPH, the joint network of ESLs and HMLs can accurately retrieve the rain rates of the real rainfall fields. In particular, the error and correlation of the RBF neural network in retrieving the rain rates from the real rainfall field are better than those of the OK algorithm. However, the performance of the RBF neural network in retrieving the average rain rate is inferior to that of the OK algorithm, and the rain rates would be underestimated for retrieving extreme rainfall.

(3) The joint network of ESLs and HMLs also shows a good performance in monitoring actual rainfall event. The results for stratiform rainfall and convective rainfall retrieved by the joint network based on the OK algorithm and the RBF neural network are substantially consistent with the distribution of actual rainfall events, and they show correctly the characteristics of stratiform rainfall and convective rainfall. Moreover, the approach of the RBF neural network performs better for the retrieval of actual rainfall events.

The paper only provides a preliminary verification of the feasibility of retrieving rainfall fields by combining ESLs and HMLs through a simulated network. With the rapid development of communication satellite constellations and 5G communication networks, the use of widely distributed networks of ESLs and HMLs to retrieve rainfall over large areas has great potential for obtaining high-precision rainfall fields and complementing traditional instruments of rainfall measurement. Next, we will build multiple real ESLs and HMLs in different topographic areas to form a rainfall observation network and monitor rainfall over a large area in real time. Furthermore, it is worthwhile to develop a study to combine ESLs and HMLs with traditional rain instruments to obtain rainfall fields with higher precision.

**Author Contributions:** Conceptualization, Y.Z. and X.L.; methodology, Y.Z. and X.L.; software, Y.Z. and M.X.; validation, X.L. and K.P.; formal analysis, J.Y. and M.X.; investigation, X.L. and J.Y.; resources, Y.Z. and X.L.; data curation, Y.Z. and X.L.; writing—original draft preparation, Y.Z.; writing—review and editing, X.L. and K.P.; visualization, Y.Z. and J.Y.; supervision, X.L.; project administration, X.L.; funding acquisition, X.L. All authors have read and agreed to the published version of the manuscript.

**Funding:** This research was funded by the National Natural Science Foundation of China (Grant No. 41975030), Excellent Youth Scholars of Natural Science Foundation of Hunan Province of China (Grant No. 2021JJ20046) and China Postdoctoral Science Foundation (Grant No. 2021M701650).

**Data Availability Statement:** The data used in this study can be obtained from the corresponding author of this article. Additionally, the data used in the simulation experiment on rainfall fields retrieved are provided by Climate Prediction Center of NOAA, which can be download from an open-access source: https://ftp.cpc.ncep.noaa.gov/precip/CMORPH_V1.0/ (accessed on 11 January 2021).

**Acknowledgments:** The authors thank the editor and anonymous reviewers for providing helpful advice.

**Conflicts of Interest:** The authors declare no conflict of interest.

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
