# Peer review of "Research on the Method of Rainfall Field Retrieval Based on the Combination of Earth–Space Links and Horizontal Microwave Links"

_remotesensing, doi:10.3390/rs14092220_

Round 1
Reviewer 1 Report
It is a well-written manuscript. However, I still have some questions for clarification. Please see the attached pdf for further details.

Author Response
Thank you for your comments on the paper, which are all valuable and very helpful for revising and improving our paper, as well as the important guiding significance to our researches. We have modified the manuscript according to your comments. The detailed responses see the attached pdf.
We appreciate for your work and hope that the corrections will meet with approval. Once again, thank you very much for your comments and suggestions.
Yingcheng Zhao
2022/1/5

Reviewer 2 Report
Very clear presentation. Only two minor improvements are suggested:
- Figure 7 – what is on the abscissa?
- Page 16, lines 500-508. Maybe it would be of interest to derive the theoretical distributions for PDF or DCF. However, it is optional.
Author Response
Response to Reviewer 2
Thank you for your comments on the paper, which are all valuable and very helpful for revising and improving our paper, as well as the important guiding significance to our researches. We have modified the manuscript according to your comments. The detailed responses are as follows.
Comments and Suggestions for Authors
Very clear presentation. Only two minor improvements are suggested:
- Figure 7: what is on the abscissa?
Response: We are very sorry for this error. The abscissa indicates the serial number of the grid points in the area, and we have redrawn the figure 7 (Figure 8 of the revised manuscript) and labelled the corresponding content.
- Page 16, lines 500-508. Maybe it would be of interest to derive the theoretical distributions for PDF or CDF. However, it is optional.
Response: Thank you very much for your suggestion. The PDF and CDF are the results of statistics on all rain rates of 300 real rainfall fields from 2017 to 2019. The theoretical distribution of statistics for more historical data on rain rates is important for the study of the raindrop size distribution (DSD) and the changes in climate in the area. We only briefly research the distribution of rain rates from 2017 to 2019 for error analysis in this paper. However, the theoretical distribution of rain rates over the area will be investigated in more depth in our other work.
We appreciate for your work and hope that the corrections will meet with approval. Once again, thank you very much for your comments and suggestions.
Yingcheng Zhao
2022/1/5
Reviewer 3 Report
This paper analyzes rainfall fields coupling with the atmospheric propagation model of Earth-Space Links and Horizontal Micro-wave links. Two simulation methods of OK algorithm and RBF neural network were used to reconstruct the sparse distribution of rainfall and a continuous distribution of rainfall fields in the area were obtained through interpolation techniques.
Introduction is well-written with relevant background information
L 95: machine learning technique
L 315: Space between number and letter: 118 °E ~ 118.35 °E and 24.35 °N ~ 24.7 °N.
L 318-319: Space between number and unit i.e. 2 km
L 353: What represents the blue markers?
L 75 – 80: Define “rainfall field”
L 317: Each grid meaning 35 km x 35 km?
L 395: the results show three rain cells. However, in the methodology it is not specify the number of rain cells (L 381)
L 381-382 Is that cell size 35× 35 km?.
L 436: Label X axis of the bottom graphs of Fig. 7. What’s the unit?
L 531 – 534: In Figure 11 caption that indicates 0.5 m hour interval for a-d, if it is applicable to e-h, and i-l, please mention it in the Figure caption.
Journal titles for some references are not abbreviated [15, 16, 27,37,45,46]
Author Response
Thank you for your comments on the paper, which are all valuable and very helpful for revising and improving our paper, as well as the important guiding significance to our researches. We have modified the manuscript according to your comments. The detailed responses can be found in the attached PDF.
